# Starjob: Dataset for LLM-Driven Job Shop Scheduling

## Abstract

Large Language Models (LLMs) have shown remarkable capabilities across various domains, but their potential for solving combinatorial optimization problems remains largely unexplored. In this paper, we investigate the applicability of LLMs to the Job Shop Scheduling Problem (JSSP), a classic NP-hard challenge. We introduce Starjob, the first large-scale supervised dataset for JSSP, comprising 130,000 instances with natural language representations designed specifically for training LLMs. Leveraging this dataset, we fine-tune a Llama-3.1 8B model using the resource-efficient RsLoRA method to create an end-to-end scheduler. Our evaluation on standard benchmarks demonstrates that this LLM-based method surpasses traditional Priority Dispatching Rules (PDRs) and achieves significant performance gains over foundational neural approaches like L2D and RASCL, with an average gap improvement of 17.36% on DMU and 7.85% on Taillard benchmarks relative to L2D. These results highlight the untapped potential of fine-tuned LLMs in combinatorial optimization, establishing a new direction for developing interactive and high-performance scheduling systems.

## 1 Introduction

Despite their success in natural language processing, Large Language Models (LLMs) have not been traditionally considered strong candidates for solving computationally intensive problems. Their applicability to NP-hard combinatorial optimization problems is often viewed as limited, a perception reinforced by the scarcity of empirical evidence showing LLMs outperforming specialized methods like reinforcement learning in these domains. Furthermore, the propensity of LLMs to "hallucinate" can lead to infeasible solutions, making their direct application unreliable. Consequently, the systematic exploration of fine-tuned LLMs for hard combinatorial problems has remained limited.

In this paper, we challenge this prevailing view by demonstrating that representation is the key to unlocking the scheduling capabilities of LLMs. We present the first fine-tuned LLM for the Job Shop Scheduling Problem (JSSP). Our results show that when trained on a properly structured, text-based representation of the problem, an LLM can not only generate feasible schedules but also outperform classic Priority Dispatching Rules (PDRs) and foundational dedicated neural methods that first surpassed them (e.g., L2D Zhang et al. (2020) and RASCLIklassov et al. (2023)). These findings suggest that with appropriate data representation and fine-tuning, LLMs can become a competitive new paradigm for combinatorial optimization, complementing existing specialized solvers.

JSSP is a fundamental optimization problem with critical applications in manufacturing and logistics, where jobs must be scheduled on machines to minimize metrics like makespan ($C_{max}$). While traditional methods face scalability challenges, modern AI techniques, particularly reinforcement learning and Graph Neural Networks (GNNs), have offered promising data-driven alternatives Zhang et al. (2020); Corsini et al. (2024). Concurrently, LLMs have been explored for tasks involving structured reasoning, such as graph analysis Huang et al. (2022); Chen et al. (2024b) and planning Valmeekam et al. (2022). However, their application to the explicit, constraint-heavy domain of scheduling remains largely unexplored.

This work bridges that gap. We are the first to employ a fine-tuned LLM for end-to-end JSSP scheduling. To enable this, we introduce **Starjob**[1], a novel supervised dataset where JSSP instances

---

[1] https://github.com/starjob42/Starjob

and their solutions are framed in natural language. By fine-tuning a Llama model with the RsLoRA method Kalajdzievski (2023) on this dataset, we demonstrate on the well-known Taillard Taillard (1993) and DMU Demirkol et al. (1998) benchmarks that our approach finds high-quality solutions, surpassing both classic PDRs and the L2D neural baseline.

Our contributions are:

- We introduce **Starjob**, the first supervised dataset with 130,000 instances designed to train LLMs for JSSP using a structured natural language format.

- We present the first end-to-end JSSP scheduler based on a fine-tuned LLM, demonstrating its ability to reason over complex constraints using the Starjob dataset and the RsLoRA method.

- We conduct a rigorous evaluation against four PDRs and neural methods L2D and RASCL, showing our model's superior performance and generalization, particularly on large-scale instances with up to 1000 operations.

- Our LLM-based approach unlocks a new modality of interaction: users can query the scheduler in natural language to understand scheduling constraints or solution characteristics, significantly enhancing transparency and usability, as presented in Listing 3.

It is important to note that while our approach demonstrates competitive performance against several established baselines, we do not claim to have developed the absolute best scheduler for JSSP. Rather, this work represents the first systematic application of LLMs to end-to-end large JSSP instances, establishing a foundation for future research at the intersection of natural language processing and combinatorial optimization.

## 2 RELATED WORK

JSSP with more than two machines is proven to be NP-hard Garey et al. (1976). As a result, finding exact solutions for JSSP is generally infeasible, leading to the widespread use of heuristic and approximate methods for practical efficiency Cebi et al. (2020). Traditional approaches to solving JSSP have primarily relied on search and inference techniques developed by the constraint programming community Beck et al. (2010). These techniques effectively leverage constraints to define the relationships and limitations between jobs and resources, enabling efficient exploration of feasible solution spaces and the identification of optimal or near-optimal schedules Nowicki & Smutnicki (2005). A widely used heuristic method in real-world scheduling systems is the Priority Dispatching Rule (PDR) Zahmani et al. (2015). PDRs are simple and effective, although designing an efficient PDR is time-consuming and requires extensive domain knowledge.

Recently, approaches utilizing Deep Learning and Neural Networks have gained attention for finding promising solutions to the JSSP Bonetta et al. (2023); Zhang et al. (2020); Corsini et al. (2024). These methods can be broadly categorized into supervised learning and reinforcement learning (RL). Current research in deep reinforcement learning (DRL) is actively focused on developing advanced methods to tackle JSSP. Existing DRL methods typically represent JSSP as a Markov Decision Process (MDP) and learn a policy network based on DRL techniquesZhang et al. (2020).

Large language models (LLMs) are now being applied to a wider range of tasks beyond language processing, in areas like robotics and planning Huang et al. (2022). While there are currently no papers that directly address the scheduling of Job Shop Scheduling Problems (JSSP) using LLMs, some notable works explore the potential of LLMs in mathematical reasoning and programming Chen et al. (2023); Wei et al. (2022); Ahn et al. (2024); Yang et al. (2023). Optimization using LLMs has gained significant interest in recent years, with several works exploring their capabilities across various domains Yang et al. (2023). The ability of LLMs to understand and generate natural language has opened new possibilities for optimization tasks that were traditionally solved using derivative-based algorithms or heuristic methods Yang et al. (2023). Chen et al. (2023) evaluated LLMs' performance in mathematical problem-solving and introduced "Program of Thoughts" (PoT) prompting. Unlike Chain of Thoughts (CoT) Wei et al. (2022), which combines reasoning and computation, PoT generates reasoning as code statements and delegates computation to an interpreter. Ahn et al. (2024) surveys mathematical problems and datasets studied with LLMs, analyzing their

strengths and weaknesses. Frieder et al. (2024) examines LLMs' impact on mathematicians, exploring their role in research, education, problem-solving, and proof generation, offering a balanced view of their capabilities. Recent works Yang et al. (2023) explore LLMs as optimizers, using prompts to refine solutions iteratively. Case studies on linear regression and the traveling salesman problem show LLMs can produce high-quality solutions, sometimes matching heuristic algorithms in small-scale scenarios. Explorations into using LLMs for graph learning tasks have yielded notable approaches. Huang et al. (2022) noted that LLMs exhibit some initial graph reasoning capabilities, but their performance decreases with problem complexity, Huang et al. (2022) introduced prompting strategies to improve LLMs graph reasoning. Valmeekam et al. (2022) developed a benchmark for assessing the planning and reasoning abilities of LLMs. More recently, Chen et al. (2024b) examined the use of LLMs for graph node classification tasks. Chen et al. (2024a) presents LLMs as enhancers for GNNs and as direct predictors from graph structures. Zhao et al. (2024) proposed GRAPHTEXT, which translates graphs to natural language for training-free reasoning, often rivaling GNNs. While LLMs show promise in graph tasks, their use in scheduling is still largely unexplored.

## 3 PRELIMINARIES

We consider the classical JSSP, which is defined as follows. Given $N_J$ jobs and $N_M$ machines, each job $J_i$ is comprised of an ordered sequence of operations $(O_{i1}, \ldots, O_{in_i})$. Each operation $O_{ij}$ must be processed on a designated machine $m_{ij}$ for a processing time $p_{ij}$. The scheduling variables $S_{ij}$ denote the start time of operation $O_{ij}$.

The JSSP is governed by two principal constraints: (i) *Precedence constraints* require that each operation starts only after the completion of its predecessor within the same job, i.e., $S_{i,j+1} \geq S_{ij} + p_{ij}$; and (ii) *Resource constraints* ensure that no two operations assigned to the same machine overlap in time, i.e., for any pair $O_{ij}, O_{kl}$ such that $m_{ij} = m_{kl}$,

$$[S_{ij}, S_{ij} + p_{ij}) \cap [S_{kl}, S_{kl} + p_{kl}) = \emptyset.$$

The objective is to minimize the makespan, defined as the maximum completion time across all operations:

$$C_{\max} = \max_{i,j}\{S_{ij} + p_{ij}\}.$$

## 4 DATASET CONSTRUCTION AND REPRESENTATION

The use of LLMs for combinatorial optimization requires translating traditional mathematical representations into natural language encodings that maintain the problem structure for language-based processing. We present a methodology for representing JSSP instances as structured natural language, mapping the conventional matrix-based form (see Listing 1) into explicit descriptions of all constraints and requirements. This mapping is defined as a deterministic, bijective transformation $\mathcal{T} : P \to \mathcal{L}$, where $P$ denotes the space of standard JSSP instances and $\mathcal{L}$ the corresponding space of natural language descriptions.

Listing 1: Example: Natural language encoding of a JSSP instance with $N_J = 3$ and $N_M = 3$.

```
Optimize the schedule for 3 jobs (J0, J1, J2) across 3 machines (
    M0, M1, M2) to minimize the makespan. Each machine can process
    only one job at a time and jobs are non-preemptive.

J0: M0:105, M1:29, M2:213
J1: M0:193, M1:18, M2:213
J2: M0:78,  M1:74, M2:221
```

For a JSSP instance with $N_J$ jobs and $N_M$ machines, we construct a natural language encoding that systematically specifies:

1. The problem dimensions ($N_J \times N_M$)

2. The operational constraints (non-preemption, machine exclusivity)

3. The sequential processing requirements for each job

4. The corresponding processing durations

This encoding establishes a bijective mapping between mathematical and linguistic representations, preserving all information required for solution generation while rendering the problem interpretable to language models. As illustrated in Listing 1, the natural language encoding presents the problem parameters in a clear, structured format that delineates job requirements across machines.

### 4.1 CORPUS GENERATION FOR MODEL TRAINING

To facilitate effective learning of the mapping between problem instances and their solutions, we constructed a comprehensive corpus of JSSP instances and their corresponding optimal or near-optimal solutions. The corpus encompasses approximately 130,000 random JSSP instances spanning dimensions from $2 \times 2$ to $20 \times 20$, supplemented by $\sim$1,000 larger and asymmetric instances to enhance generalization capabilities across problem scales. Operation durations were sampled from a uniform distribution ranging from 5 to 500 time units, ensuring comprehensive coverage of the solution space and robustness to varying temporal constraints. The testing dataset is out of distribution dataset from the trainng dataset. We conduct evaluations on the TAI Taillard (1993) and DMU Demirkol et al. (1998) benchmark sets, which are entirely held out from the training phase.

For solution generation, each instance was processed using Google's OR-Tools optimization framework with parameters configured to balance computational efficiency and solution quality. The solver was allocated a 300-second time limit with 42 parallel workers utilizing the AUTOMATIC_SEARCH strategy, providing near-optimal solutions even for larger problem instances. For problems exceeding $10 \times 10$ dimensions, we acknowledge potential suboptimality due to computational constraints while maintaining solution feasibility.

The solution encoding adopts a structured natural language format, specifically designed to guide the autoregressive nature of the LLMs. Each entry in the solution sequence (see Listing 2) details a job-machine assignment along with its explicit start time, duration, and resulting completion time. Notably, the use of summation notation (e.g., "J2-M0: 0+78 → 78") forces the model to compute the current makespan incrementally, based on the start time and duration, while taking into account the completion times of all previously scheduled operations.

This stepwise representation leverages the LLM's autoregressive generation process, requiring it to "think" about the current scheduling decision by explicitly calculating and verifying the timing constraints before proceeding to the next operation. The format ensures that each scheduling step is conditioned on the already constructed partial schedule, thus embedding temporal dependencies and constraint satisfaction directly into the generation process.

Listing 2: Consistent schedule with correct job precedence and operation durations for a JSSP instance with $N_J = 3$ jobs and $N_M = 3$ machines. The values after "-¿" denote operation completion times. The makespan is the maximum of these, i.e., $488$.

```
    Solution:
J2-M0: 0+78 -> 78,
J1-M2: 0+193 -> 193,
J0-M0: 78+105 -> 183,
J0-M1: 183+29 -> 212,
J2-M2: 193+74 -> 267,
J1-M1: 212+18 -> 230,
J1-M0: 230+213 -> 443,
J2-M1: 267+221 -> 488,
J0-M2: 267+213 -> 480
```

Empirical results (see Table 3) show that this explicit, computation-driven format significantly enhances the feasibility of generated solutions compared to formats omitting intermediate calculations. By prompting the model to perform and record intermediate makespan computations, the approach

enables real-time constraint checking and more effective optimization, reducing the frequency of infeasible schedules.

## 5 METHODOLOGY

We propose a novel method for solving JSSP by fine-tuning large language models with natural language representations of scheduling problems and solutions. Our framework, based on Meta-Llama-3.1-8B-Instruct (4-bit quantized), reframes JSSP as a sequence generation task and operates in two phases: (1) fine-tuning the model on problem-solution pairs using rsLoRA, and (2) generating and selecting optimal schedules for new instances. By expressing both problems and solutions in natural language, our approach leverages pre-trained knowledge and learns scheduling-specific patterns efficiently.

### 5.1 TRAINING METHODOLOGY

We fine-tune the model using rsLoRA Hu et al. (2022), an approach that replaces the standard scaling factor $\frac{\alpha}{r}$ with a stabilized $\sqrt{\frac{\alpha}{r}}$, which enables the use of higher ranks without causing gradient collapse and ensures more stable training dynamics. The model is initialized with pre-trained weights $\theta_0$, which remain frozen throughout the process, while only the low-rank adaptation matrices $U$ and $V$ are updated to minimize the negative log-likelihood loss on tokenized problem-solution pairs. Training is conducted over 2 epoch with a learning rate of $2 \times 10^{-4}$, LoRA rank $r = 64$, scaling factor $\alpha = 64$, and a batch size of 16, utilizing a single Nvidia RTX A6000 GPU (48GB memory), with the training process taking approximately 70 hours and utilizing around 30GB of GPU RAM, highlighting the resource requirements for this procedure. Alpaca training template format is used during the training as described in Appendix Listings 1.

---

**Algorithm 1:** LLM Fine-Tuning for JSSP with rsLoRA

**Input:** Problem instance $\mathcal{L}_p$ in natural language, Fine-tuned LLM with parameters
$\theta = \theta_0 + \gamma_r U V^\top$ , Number of candidate solutions $S$

1 Initialize low-rank matrices $U, V \in \mathbb{R}^{d \times r}$
2 Define rank-stabilized factor $\gamma_r = \frac{\alpha}{\sqrt{r}}$
3 **for** *epoch* = 1 **to** $E$ **do**
4    **for** *each batch* $\{(\mathcal{L}_p^{(i)}, s^{(i)})\}_{i=1}^B \subset \mathcal{D}$ **do**
5       Tokenize each problem $\mathcal{L}_p^{(i)}$ and solution $s^{(i)}$
6       Construct inputs with problems $\mathcal{L}_p^{(i)}$ as context
7       Set targets as tokenized solutions $s^{(i)} = \{w_1^{(i)}, \dots, w_{T_i}^{(i)}\}$
8       Compute effective parameters: $\theta = \theta_0 + \gamma_r U V^\top$
9       Forward pass: Compute probabilities $p(w_t | w_{<t}, \mathcal{L}_p^{(i)}; \theta)$
10      Compute NLL loss: $\mathcal{L} = -\sum_{i=1}^B \sum_{t=1}^{T_i} \log p(w_t^{(i)} | w_{<t}^{(i)}, \mathcal{L}_p^{(i)}; \theta)$
11      Compute gradients $\nabla_U \mathcal{L}$ and $\nabla_V \mathcal{L}$
12      Update $U \leftarrow U - \eta \nabla_U \mathcal{L}$
13      Update $V \leftarrow V - \eta \nabla_V \mathcal{L}$
14    Evaluate model performance on validation set

**Result:** Fine-tuned model parameters $\theta = \theta_0 + \gamma_r U V^\top$

---

### 5.2 INFERENCE AND SOLUTION SELECTION

At inference time (Algorithm 2), we employ a generate-and-select strategy. For each problem instance, the model produces multiple candidate solutions through temperature-controlled sampling. Each candidate undergoes rigorous feasibility checking to ensure satisfaction of all JSSP constraints, including job precedence, machine exclusivity, and non-preemption requirements. From the set of feasible solutions, we select the one with the minimum makespan.

The feasibility check validates that: (1) each job's operations are scheduled in the correct sequence, (2) no machine processes multiple jobs simultaneously, (3) each operation has the correct processing time, and (4) all operations are scheduled exactly once. This comprehensive validation ensures that all solutions adhere to the fundamental constraints of the JSSP problem domain.

---

**Algorithm 2:** LLM-Based JSSP Solution Generation and Selection

**Input:** Problem instance $\mathcal{L}_p$ in natural language, Fine-tuned LLM with parameters
$\quad\quad \theta = \theta_0 + \gamma_r UV^\top$ ,Number of candidate solutions $S$, temperature $\tau$

1   Initialize empty set of feasible solutions $\mathcal{S}_p^f \leftarrow \emptyset$

2   **for** $i = 1$ **to** $S$ **do**

3      Generate candidate solution $s_i \sim \text{LLM}_\theta(\mathcal{L}_p, \tau)$

4      Parse solution $s_i$ to extract job-machine assignments and timings

5      $\text{valid}_{\text{precedence}} \leftarrow$ Check job operation precedence constraints

6      $\text{valid}_{\text{exclusivity}} \leftarrow$ Check machine exclusivity constraints

7      $\text{valid}_{\text{timing}} \leftarrow$ Check correct processing times

8      $\text{valid}_{\text{completeness}} \leftarrow$ Check all operations are scheduled once

9      **if** $valid_{precedence} \wedge valid_{exclusivity} \wedge valid_{timing} \wedge valid_{completeness}$ **then**

10        Compute makespan $M(s_i)$

11        Add to feasible set: $\mathcal{S}_p^f \leftarrow \mathcal{S}_p^f \cup \{s_i\}$

12   **if** $\mathcal{S}_p^f \neq \emptyset$ **then**

13      **return** $s^* = \arg\min_{s \in \mathcal{S}_p^f} M(s)$        // Best solution by makespan

14   **else**

15      **return** "No feasible solution found"

---

## 6   Validation, Baseline Methods, and Empirical Analysis

We evaluated our end-to-end LLM-based job shop scheduler using the standard Taillard Taillard (1993) and DMU Demirkol et al. (1998) benchmarks, comparing it to both traditional heuristics and state-of-the-art learning-based methods. As the first application of LLMs for end-to-end JSSP solution generation, we benchmarked our model against L2D Zhang et al. (2020)—an early neural approach that outperforms classic priority dispatching rules (PDRs) such as SPT, MWKR, MOPNR, and FDD/MWKR. L2D leverages a graph neural network and PPO Schulman et al. (2017) for generalization. Additionally, we compared our method with RASCLB Iklassov et al. (2023), a state-of-the-art reinforcement learning approach designed for cross-instance generalization. Here, "B" denotes the "base" learning method in Iklassov et al. (2023), which combines an RL-based method with rLSTM and set2set modules. RASCLB is trained on larger instances (30x20) with a sample size of 20. Its reverse LSTM Hochreiter & Schmidhuber (1997) component receives static, multidimensional embeddings for all operations in a job $J_i$, propagating information backward from the last operation to the current one. For all experiments, inference was performed with a context length of 40,000 tokens (the maximum number of tokens the model can process in a single input sequence) using default sampling settings and $S = 20$ samples per instance, with default temperature parameter of 1. Both training and inference used `4-bit` quantization for memory efficiency, requiring about 30GB of GPU memory on an NVIDIA A6000. Our largest evaluated instance (23,000 tokens) fits comfortably within this window. For faster inference, we converted the model to the `llama.cpp` format Gerganov (2023), achieving 102.22 tokens/sec on an RTX A6000 (48GB), as reported by Dai et al. Dai (2024). Notably, inference time scales with token sequence length rather than problem complexity; processing our largest instance (22,224 tokens) within the 40,000-token window took about 217 seconds per sample, regardless of task type.

### 6.1   Performance Metrics and Comparative Results

Performance on each benchmark was evaluated using the *Percentage Gap* (PG), defined as:

$$\text{PG} \;=\; 100 \times \left( \frac{M_{\text{alg}}}{M_{\text{ub}}} \;-\; 1 \right),$$

where $M_{\text{alg}}$ is the makespan produced by the algorithm, and $M_{\text{ub}}$ is the best-known or optimal makespan. Lower PG values correspond to solutions closer to the best-known objective. Tables 1 and 2 present the gap percentages of the fine-tuned Llama model compared to L2D and other methods (SPT, MWKR, FDD/WKR, MOPNR) on Tai and DMU datasets. Benchmark solutions are available at[2] and[3]. Across instances ranging from 15 jobs and 15 machines to 50 jobs and 20 machines, the fine-tuned Llama 3.1 consistently outperforms all other methods. On larger instances (50 jobs, 20 machines, 1000 nodes), Llama achieves a 24.32% gap, surpassing L2D (26.40%). The average gap for fine-tuned Llama (21.69%) is significantly lower than SPT (60.57%), MWKR (55.29%), FDD/WKR (46.77%), MOPNR (42.99%), and L2D (29.54%). On the DMU benchmark with 50 jobs and 15 machines, fine-tuned Llama (22.14%) again outperforms all methods, including L2D (37.50%). SPT consistently has the highest gap (often above 60%), reflecting its inability to handle complex constraints. MWKR and FDD/WKR, while more sophisticated, remain outperformed by learning-based approaches due to their myopic decision processes. Notably, our method shows average improvements over the dedicated model RASCLB of 13.41% on the Taillard dataset (see Table 1) and comparableresult on the DMU dataset (see Table 2).

## 6.2 ABLATION STUDY

To understand the impact of key components, we conducted an ablation study focusing on our **Starjob** data representation and the model size. As shown in Table 3, the our proposed summation format significantly increases the percentage of feasible solutions produced by the Llama 8B model compared to the standard matrix format, not involving summation operations, which is familiar to the LLM from pretraining Bordt et al. (2024). Despite this familiarity, Starjob yields markedly better feasibility. The Llama 1B model, in contrast, failed on this task, highlighting the necessity of larger models for high-quality scheduling.

---

Listing 3: An interactive session where the LLM explains the primary scheduling bottleneck.

> **Example: Interactive Query with the Starjob-LLM**
>
> **User:**
> Looking at the 3x3 problem instance, what was the main bottle-
> neck that made it hard to solve?
>
> **Starjob-LLM:**
> The primary bottleneck was **Machine 2 (M2).** All three jobs
> required this machine for long processing times, creating a
> highly contended resource.
> The sequential processing on M2 (J1-M2 → J2-M2 → J0-M2)
> formed the critical path, which ultimately determined the fi-
> nal makespan of 488. Any improvement to the schedule would
> need to resolve this contention on Machine 2.

---

## 7 CONCLUSION

This work served as a test to see whether LLMs could generate any feasible solutions for NP-hard combinatorial optimization, rather than to claim the best scheduler. We showed that, with the right data representation, LLMs can indeed act as effective schedulers. To this end, we introduced **Starjob**, a large supervised dataset for the Job Shop Scheduling Problem (JSSP) in structured natural language, and fine-tuned a Llama 8B model using resource-efficient methods. While our goal was not to surpass specialized schedulers, our LLM-based approach still outperformed traditional Priority Dispatching Rules and even some dedicated neural baselines on standard benchmarks. These findings highlight the potential of LLMs for combinatorial optimization, and suggest promising directions for more interpretable and interactive scheduling systems in the future.

---

[2] http://optimizizer.com/TA.php
[3] http://jobshop.jjvh.nl/

## 8 LIMITATIONS AND FUTURE WORK

While our model's performance is dependent on the Starjob dataset and constrained by the LLM's context window, it provides high-quality heuristic solutions. Future work will focus on developing hybrid solvers that combine our approach with traditional optimization methods for further refinement. We also plan to explicitly integrate Graph Neural Networks (GNNs) into the LLM latent space via cross-attention to better capture relational structure, and to investigate the impact of full fine-tuning and larger LLMs for additional performance gains.

Table 1: Comparison of different methods on the **TAI** dataset (sampling budget = 20). Lower values indicate schedules closer to the optimal solution, representing better performance. * indicates the best result according to the Percentage Gap. Classic JSSP heuristics (FDD/WKR, MOPNR, MWKR, SPT) are described in Appendix D. *L2D*, *RASCLB*, and *LLM-FT-Ours* are neural methods.

| Method | 15x15 | 20x15 | 20x20 | 30x15 | 30x20 | 50x15 | 50x20 | Average |
|---|---|---|---|---|---|---|---|---|
| FDD/WKR | 47.45 | 50.57 | 47.57 | 45.01 | 56.30 | 37.72 | 42.80 | 46.77 |
| MOPNR | 44.98 | 47.97 | 43.68 | 45.59 | 48.23 | 31.25 | 39.24 | 42.99 |
| MWKR | 56.74 | 60.65 | 55.60 | 52.61 | 63.93 | 41.90 | 55.62 | 55.29 |
| SPT | 54.64 | 65.24 | 64.11 | 61.61 | 66.03 | 51.37 | 61.00 | 60.57 |
| L2D | 25.95 ± 3.37 | 30.03 ± 3.90 | 31.60 ± 4.11 | 33.02 ± 4.29 | 33.62 ± 4.37 | 26.15 ± 3.40 | 26.40 ± 3.43 | 29.54 ± 3.84 |
| RASCLB | 20.59 ± 2.47 | 25.31 ± 3.04 | 25.47 ± 3.06 | 27.27 ± 3.27 | 30.40 ± 3.65 | 20.69 ± 2.48 | 26.40 ± 3.17 | 25.16 ± 3.02 |
| LLM-FT-Ours | **19.34 ± 1.93*** | **18.00 ± 1.80*** | **21.11 ± 2.11*** | **21.44 ± 2.14*** | **30.05 ± 3.00*** | **17.57 ± 1.76*** | **24.32 ± 2.43*** | **21.69 ± 2.17*** |

Table 2: Comparison of different methods on the **DMU** dataset (sampling budget = 60). Lower values indicate schedules closer to the optimal solution, representing better performance. * indicates the best result according to the Percentage Gap. Classic JSSP heuristics (FDD/WKR, MOPNR, MWKR, SPT) are described in Appendix D. *L2D*, *RASCLB*, and *LLM-FT-Ours* are neural methods.

| Method | 20x15 | 20x20 | 30x15 | 30x20 | 40x15 | 40x20 | 50x15 | Average |
|---|---|---|---|---|---|---|---|---|
| FDD/WKR | 53.58 | 52.51 | 54.12 | 60.08 | 50.76 | 55.52 | 37.58 | 52.02 |
| MOPNR | 49.17 | 45.18 | 47.14 | 51.97 | 43.23 | 49.22 | 31.73 | 45.38 |
| MWKR | 62.14 | 58.16 | 60.96 | 63.15 | 52.40 | 61.09 | 43.23 | 57.30 |
| SPT | 64.12 | 64.55 | 62.57 | 65.92 | 55.89 | 62.99 | 47.83 | 60.55 |
| L2D | 38.95 ± 5.06 | 37.74 ± 4.91 | 41.86 ± 5.44 | 39.48 ± 5.13 | 36.68 ± 4.77 | 41.18 ± 5.35 | 26.60 ± 3.46 | 37.50 ± 4.88 |
| RASCLB | 19.66 ± 2.36 | **15.98 ± 1.92** | **16.35 ± 1.96** | 23.00 ± 2.76 | 17.89 ± 2.15 | 26.42 ± 3.17 | 21.84 ± 2.62 | 20.16 ± 2.42 |
| LLM-FT-Ours | **19.20 ± 1.92*** | 20.16 ± 2.02 | 22.11 ± 2.21 | **21.82 ± 2.18*** | **17.24 ± 1.72*** | **23.61 ± 2.36*** | **16.85 ± 1.69*** | **20.14 ± 2.01*** |

Table 3: Ablation study comparing our **Starjob** representation against the standard **Matrix** format, and the Llama 8B model against Llama 1B model. Average Feasibility (%) indicates the percentage of valid solutions generated, where **higher values are better**. Our proposed format dramatically increases feasibility. The complete failure of the Llama 1B model highlights the task's complexity, while the Llama 8B model using Starjob consistently produces high-quality schedules. Feasibility and time for Llama 1B model are marked as N/A where not available.

| Problem Size | **Without** Summation (%) | | **With** Summation (%) | | Avg. Time (s) (8B) |
|---|---|---|---|---|---|
| | 1B | 8B | 1B | 8B | |
| 5×5 | N/A | ∼8.0 | N/A | ∼9.5 | 6.1 |
| 8×8 | N/A | ∼8.0 | N/A | ∼10.5 | 7.2 |
| 10×10 | N/A | ∼1.0 | N/A | ∼12.1 | 2.6 |
| 12×12 | N/A | ∼4.0 | N/A | ∼39.6 | 14.3 |
| 15×15 | N/A | ∼1.0 | N/A | ∼14.4 | 22.5 |
| 20×20 | N/A | ∼1.5 | N/A | ∼17.4 | 15.1 |
| 30×30 | N/A | N/A | N/A | ∼30.5 | 18.3 |
| 50×20 | N/A | ∼1.0 | N/A | ∼14.6 | 22.0 |

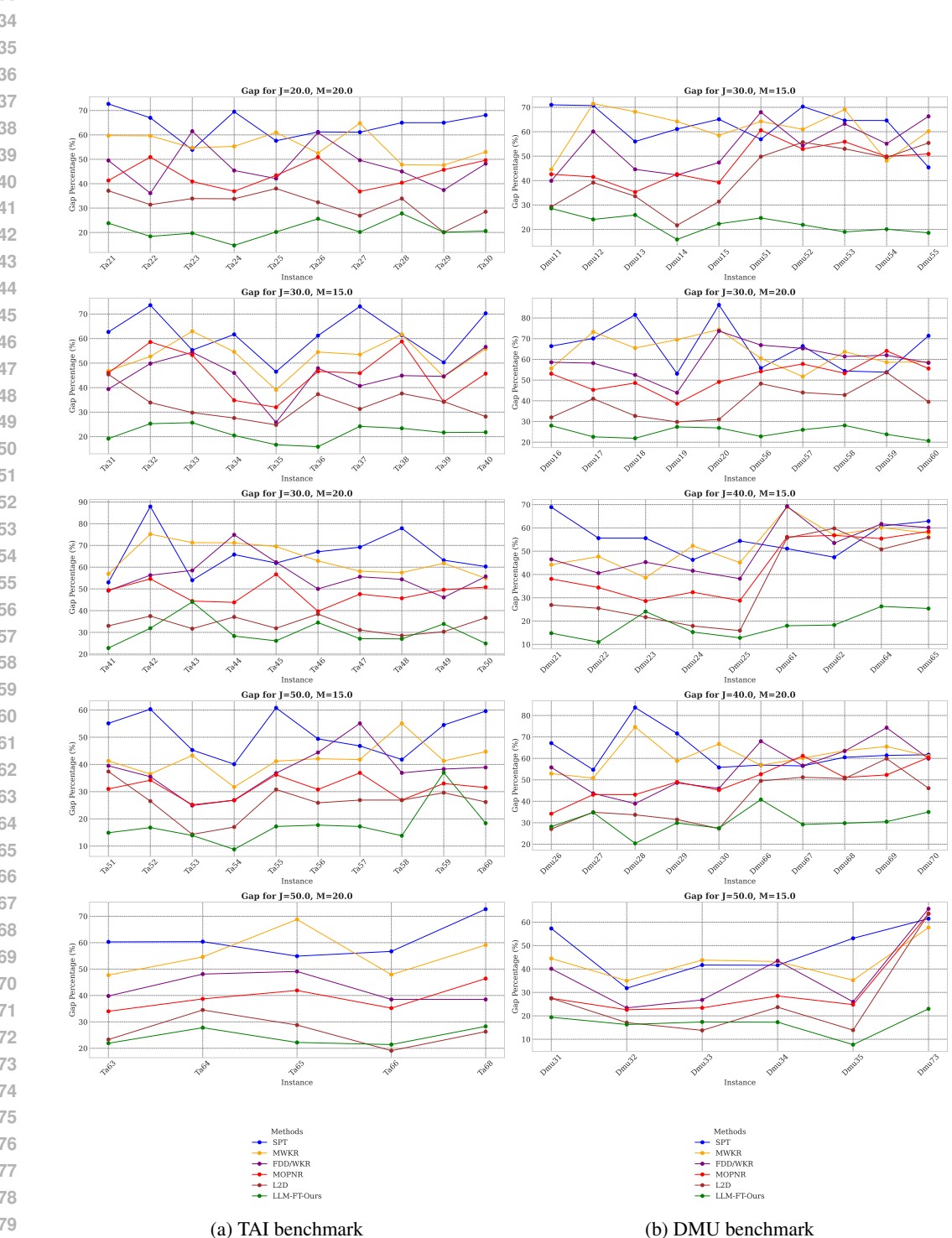

(a) TAI benchmark        (b) DMU benchmark

Figure 1: Comparison of different methods on TAI Taillard (1993) and DMU Demirkol et al. (1998).

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

## A  APPENDIX

## B  TRAINING DETAILS

### MODEL OVERVIEW

The model being fine-tuned is LLaMA 3.1, an 8 billion parameter model from MetaAI (2024a), using a 4-bit quantized version to reduce memory usage. Finetuning was conducted using Stabilized Low-Rank Adaptation (RsLoRA) with rank $r = 64$ to introduce learnable parameters specifically in

targeted layers. Kalajdzievski (2023) Compared to LoraHu et al. (2022) RsLoRa improves the stability of training by modifying the rank during adaptationKalajdzievski (2023). The target modules include:

$$\text{target\_modules} = \{\texttt{q\_proj, k\_proj, v\_proj, o\_proj,}$$
$$\texttt{gate\_proj, up\_proj, down\_proj}\} \quad (1)$$

The LoRA-specific parameters are configured as follows:

- Rank ($r$): 64
- LoRA Alpha ($\alpha$): 64
- LoRA Dropout: 0
- Bias: none

This resulted in number of trainable parameters $= 167,772,160$ or 2 % of the entire Llama 8B model's parameters.

## QUANTIZATION AND MEMORY EFFICIENCY

The model is loaded in 4-bit precision to reduce memory consumption. Gradient checkpointing is enabled using the `unsloth` AI (2024b) method, to fit longer sequences by saving memory. This reduces the VRAM usage by approximately 30%, enabling larger batch sizes.

## TRAINING PARAMETERS

The fine-tuning process is controlled by the following parameters:

- **Batch size per device**: 4
- **Gradient accumulation steps**: 4
- **Max sequence length**: 40,000 tokens
- **Number of epochs**: 2
- **Warmup steps**: 5
- **Learning rate**: $2 \times 10^{-4}$
- **Optimizer**: AdamW with 8-bit precision
- **Weight decay**: 0.01
- **Learning rate scheduler**: Linear decay
- **FP16 precision**:True

Listing 1: Prompt format used during training.

```
alpaca_prompt = """Below is an instruction that describes a task,
paired with an input that provides further context.
Write a response that appropriately completes the request.

    ### Instruction:
    {}

    ### Input:
    {}

    ### Response:
    {}"""
```

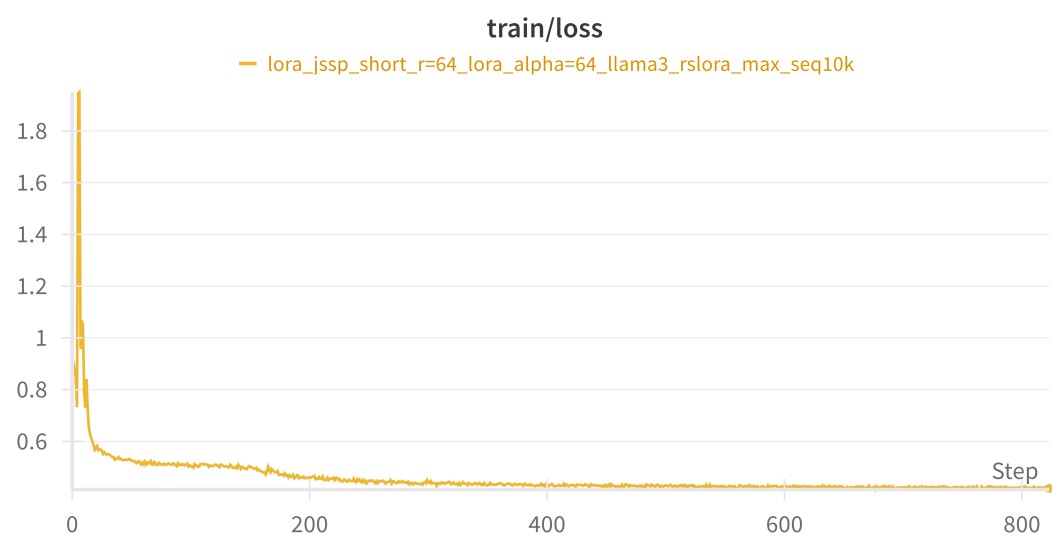

Figure 2: Train loss of Llama 8B 4bt model on Starjob dataset

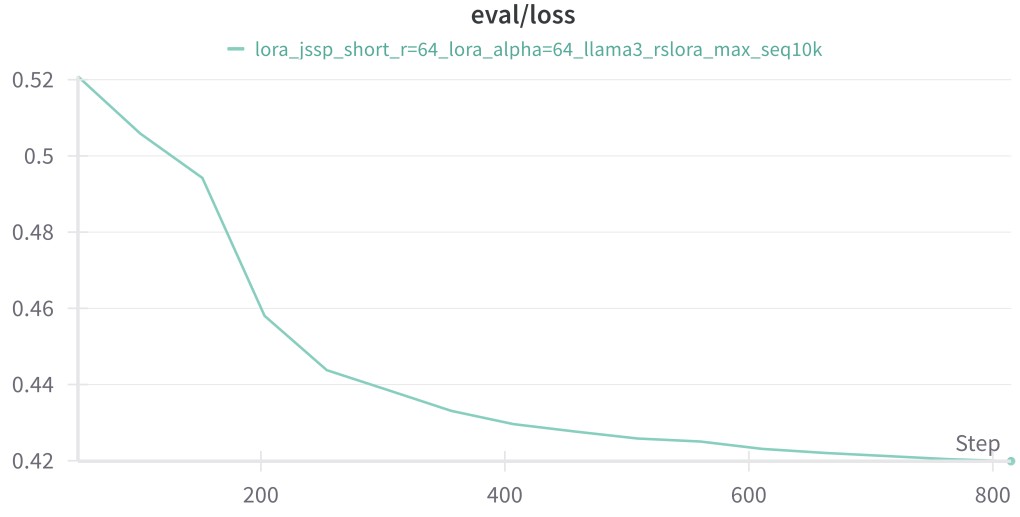

Figure 3: Eval Loss of Llama 8B 4bt model on Starjob dataset

## DATA AND DATASET SPLITTING

The dataset used for training is a local version of the proposed Starjob dataset, and it is split into 98% training and 2% evaluation:

$$\text{train : eval} = 98\% : 2\%$$

The prompts are formatted using a predefined Alpaca-style template, which ensures the model is trained on instruction-following tasks.

## EVALUATION AND SAVING STRATEGY

The best model was loaded at the end of training based on the evaluation loss:

$$\text{Metric for Best Model} = \text{Evaluation Loss}$$

Total number of saved models is limited to 50 to prevent excessive memory usage.

## GPU UTILIZATION

The training process takes place on Nvidia A6000 GPU with 48GB of memory. Training took around 70 hours and required 30GB of GPU RAM.

## C  GENERAL STATISTICS ABOUT DATASET

The dataset comprises 130,000 randomly generated JSSP instances with solutions in natural language, provided in `.json` format with the following columns:

- `num_jobs` (int64): 12 unique values.
- `num_machines` (int64): 12 unique values.
- `instruction` (object): 130,000 unique values. Initial problem description detailing jobs and machines.
- `input` (object): 130,000 unique values. Problem description formatted for LLM.
- `output` (object): 130,000 unique values. Solution in LLM format.
- `matrix` (object): 130,000 unique values. OR-Tool makespan and solution in matrix format.

The `output` column serves as the target or label column, providing the solution to the JSSP problem in natural language and the associated makespan.

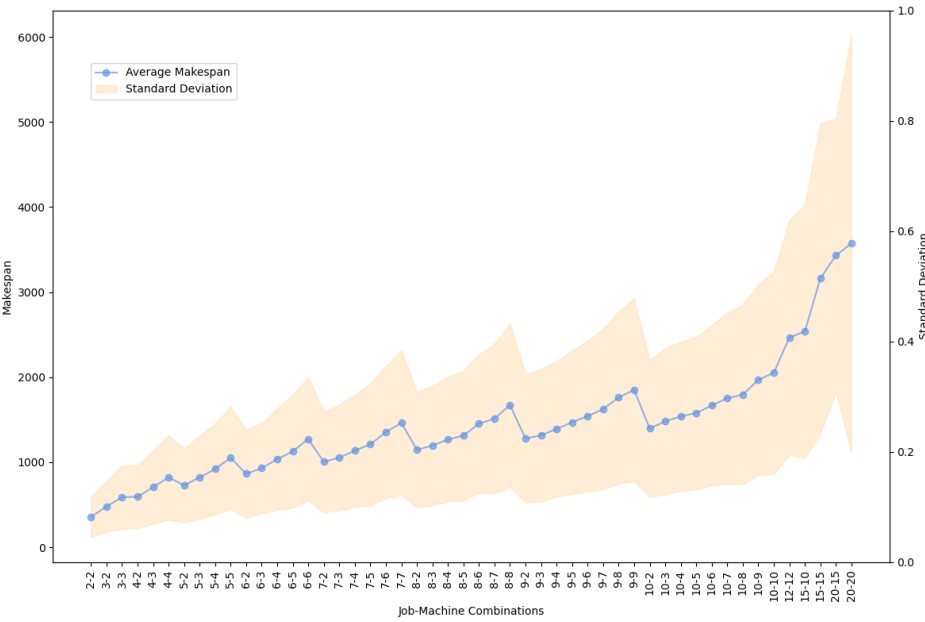

Figure 4: Makespan metrics across different job-machine combinations. The x-axis represents the combinations of jobs and machines (e.g., a 3-2 instance refers to 3 jobs and 2 machines), the right y-axis shows the standard deviation, while the left y-axis shows the makespan values.

Table 4: Comparison of PDRs against L2D against Finetuned Llama on Starjob dataset and the average Gaps on Tai Benchmark Dataset. The lower the value, the closer the schedule is to the optimal solution, thus representing better performance. BFL* indicates best from literature.

| J | M | Instance | SPT | MWKR | FDD/WKR | MOPNR | L2D | BFL* | Llama-FT-Ours |
|---|---|---|---|---|---|---|---|---|---|
| 15 | 15 | Ta01 | 1872 (52.1%) | 1786 (45.1%) | 1841 (49.6%) | 1864 (51.4%) | 1443 (17.2%) | 1231.0* | 1453.0 (18.0%) |
| 15 | 15 | Ta02 | 1709 (37.4%) | 1944 (56.3%) | 1895 (52.3%) | 1680 (35.0%) | 1544 (24.1%) | 1244.0* | 1440.0 (15.8%) |
| 15 | 15 | Ta03 | 2009 (64.9%) | 1947 (59.9%) | 1914 (57.1%) | 1558 (27.9%) | 1440 (18.2%) | 1218.0* | 1521.0 (24.9%) |
| 15 | 15 | Ta04 | 1825 (53.3%) | 1694 (44.2%) | 1653 (40.7%) | 1755 (49.4%) | 1637 (39.3%) | 1175.0* | 1387.0 (18.0%) |
| 15 | 15 | Ta05 | 2044 (67.0%) | 1892 (54.6%) | 1787 (46.0%) | 1605 (31.1%) | 1619 (32.3%) | 1224.0* | 1461.0 (19.4%) |
| 15 | 15 | Ta06 | 1771 (43.1%) | 1976 (59.6%) | 1748 (41.2%) | 1815 (46.6%) | 1601 (29.3%) | 1238.0* | 1499.0 (21.1%) |
| 15 | 15 | Ta07 | 2016 (64.3%) | 1961 (59.8%) | 1660 (35.3%) | 1884 (53.5%) | 1568 (27.8%) | 1227.0* | 1473.0 (20.0%) |
| 15 | 15 | Ta08 | 1654 (35.9%) | 1803 (48.2%) | 1839 (51.1%) | 1839 (51.1%) | 1468 (20.6%) | 1217.0* | 1475.0 (21.2%) |
| 15 | 15 | Ta09 | 1962 (54.0%) | 2215 (73.9%) | 1848 (45.1%) | 2002 (57.1%) | 1627 (27.7%) | 1274.0* | 1534.0 (20.4%) |
| 15 | 15 | Ta10 | 2164 (74.4%) | 2057 (65.8%) | 1937 (56.1%) | 1821 (46.7%) | 1527 (23.0%) | 1241.0* | 1465.0 (18.0%) |
| 20 | 15 | Ta11 | 2212 (63.0%) | 2117 (56.0%) | 2101 (54.8%) | 2030 (49.6%) | 1794 (32.2%) | 1357.0* | 1691.0 (24.6%) |
| 20 | 15 | Ta12 | 2414 (76.6%) | 2213 (61.9%) | 2034 (48.8%) | 2117 (54.9%) | 1805 (32.0%) | 1367.0* | 1677.0 (22.7%) |
| 20 | 15 | Ta13 | 2346 (74.7%) | 2026 (50.9%) | 2141 (59.4%) | 1979 (47.4%) | 1932 (43.9%) | 1343.0* | 1749.0 (30.2%) |
| 20 | 15 | Ta14 | 2190 (56.8%) | 2164 (60.9%) | 1841 (36.9%) | 2036 (51.4%) | 1664 (23.7%) | 1345.0* | 1660.0 (23.4%) |
| 20 | 15 | Ta15 | 2163 (61.5%) | 2180 (62.6%) | 2187 (63.3%) | 1939 (44.8%) | 1730 (29.2%) | 1339.0* | 1770.0 (32.2%) |
| 20 | 15 | Ta16 | 2232 (64.1%) | 2528 (85.9%) | 1926 (41.6%) | 1980 (45.6%) | 1710 (25.7%) | 1360.0* | 1731.0 (27.3%) |
| 20 | 15 | Ta17 | 2185 (49.5%) | 2015 (37.8%) | 2093 (43.2%) | 2211 (51.2%) | 1897 (29.8%) | 1462.0* | 1846.0 (26.3%) |
| 20 | 15 | Ta18 | 2267 (62.4%) | 2275 (63.0%) | 2064 (47.9%) | 1981 (44.9%) | 1794 (28.5%) | 1396.0* | 1706.0 (22.2%) |
| 20 | 15 | Ta19 | 2238 (68.0%) | 2201 (65.2%) | 1958 (47.0%) | 1899 (42.6%) | 1682 (26.3%) | 1332.0* | 1685.0 (26.5%) |
| 20 | 15 | Ta20 | 2370 (75.8%) | 2188 (62.3%) | 2195 (62.8%) | 1986 (47.3%) | 1739 (29.0%) | 1348.0* | 1802.0 (33.7%) |
| 20 | 20 | Ta21 | 2836 (72.7%) | 2622 (59.7%) | 2455 (49.5%) | 2320 (41.3%) | 2252 (37.1%) | 1642.0* | 2077.0 (26.5%) |
| 20 | 20 | Ta22 | 2672 (67.0%) | 2554 (59.6%) | 2177 (36.1%) | 2415 (50.9%) | 2102 (31.4%) | 1600.0* | 2443.0 (52.7%) |
| 20 | 20 | Ta23 | 2397 (53.9%) | 2408 (54.7%) | 2514 (61.5%) | 2194 (40.9%) | 2085 (33.9%) | 1557.0* | 2086.0 (34.0%) |
| 20 | 20 | Ta24 | 2787 (69.5%) | 2553 (55.3%) | 2391 (45.4%) | 2250 (36.9%) | 2200 (33.8%) | 1644.0* | 2135.0 (29.9%) |
| 20 | 20 | Ta25 | 2513 (57.6%) | 2582 (61.0%) | 2267 (42.1%) | 2146 (43.4%) | 2201 (38.0%) | 1595.0* | 2304 (44.4%) |
| 20 | 20 | Ta26 | 2649 (61.2%) | 2506 (52.5%) | 2484 (60.9%) | 2284 (50.9%) | 2176 (32.4%) | 1643.0* | 2195.0 (33.6%) |
| 20 | 20 | Ta27 | 2707 (61.1%) | 2768 (64.8%) | 2514 (49.6%) | 2298 (36.8%) | 2132 (26.9%) | 1680.0* | 2172.0 (29.3%) |
| 20 | 20 | Ta28 | 2654 (65.0%) | 2370 (47.8%) | 2330 (45.0%) | 2259 (40.4%) | 2146 (33.9%) | 1603.0* | 2088.0 (30.3%) |
| 20 | 20 | Ta29 | 2681 (65.0%) | 2399 (47.6%) | 2322 (37.4%) | 2367 (45.7%) | 1952 (20.1%) | 1625.0* | 2209 (35.9%) |
| 20 | 20 | Ta30 | 2662 (68.1%) | 2424 (53.0%) | 2348 (48.2%) | 2370 (49.6%) | 2035 (28.5%) | 1584.0* | 2038.0 (28.7%) |

Table 5: Comparison of PDRs against L2D against Finetuned Llama on Starjob dataset and the average Gaps on DMU Benchmark Dataset. The lower the value, the closer the schedule is to the optimal solution, thus representing better performance. BFL* indicates best from literature.

| J | M | Instance | SPT | MWKR | FDD/WKR | MOPNR | L2D | BFL* | Llama-FT-Ours |
|---|---|---|---|---|---|---|---|---|---|
| 20 | 15 | Dmu01 | 4516 (76.2%) | 3988 (55.6%) | 3535 (37.9%) | 3882 (51.5%) | 3323 (29.7%) | 2563.0* | 3064 (19.5%) |
| 20 | 15 | Dmu02 | 4593 (69.7%) | 4555 (68.3%) | 3847 (42.2%) | 3884 (43.5%) | 3630 (34.1%) | 2706.0* | 3233 (19.5%) |
| 20 | 15 | Dmu03 | 4438 (62.5%) | 4117 (50.8%) | 4063 (48.8%) | 3979 (45.7%) | 3660 (34.0%) | 2731.0* | 3296 (20.7%) |
| 20 | 15 | Dmu04 | 4533 (69.8%) | 3995 (49.7%) | 4160 (55.9%) | 4079 (52.8%) | 3816 (43.0%) | 2669.0* | 3299 (23.6%) |
| 20 | 15 | Dmu05 | 4420 (60.8%) | 4977 (81.0%) | 4238 (54.2%) | 4116 (49.7%) | 3897 (41.8%) | 2749.0* | 3458 (25.8%) |
| 20 | 15 | Dmu41 | 5283 (62.7%) | 5377 (65.5%) | 5187 (59.7%) | 5070 (56.1%) | 4316 (32.9%) | 3248.0* | 4137 (27.4%) |
| 20 | 15 | Dmu42 | 5354 (57.9%) | 6076 (79.2%) | 5583 (64.7%) | 4976 (46.8%) | 4858 (43.3%) | 3390.0* | 4169 (23.0%) |
| 20 | 15 | Dmu43 | 5328 (54.8%) | 4938 (43.5%) | 5086 (47.8%) | 5012 (45.7%) | 4887 (42.0%) | 3441.0* | 4634 (34.7%) |
| 20 | 15 | Dmu44 | 5745 (64.7%) | 5630 (61.4%) | 5550 (59.1%) | 5213 (49.5%) | 5151 (47.7%) | 3488.0* | 4429 (27.0%) |
| 20 | 15 | Dmu45 | 5305 (62.1%) | 5446 (66.4%) | 5414 (65.5%) | 4921 (50.4%) | 4615 (41.0%) | 3272.0* | 4423 (35.2%) |
| 20 | 20 | Dmu06 | 6230 (92.0%) | 5556 (71.3%) | 5258 (62.1%) | 4747 (46.3%) | 4358 (34.3%) | 3244.0* | 4173 (28.6%) |
| 20 | 20 | Dmu07 | 5619 (84.5%) | 4636 (52.2%) | 4789 (57.2%) | 4367 (43.4%) | 3671 (20.5%) | 3046.0* | 3821 (25.4%) |
| 20 | 20 | Dmu08 | 5239 (64.3%) | 5078 (59.3%) | 4817 (51.1%) | 4480 (40.5%) | 4048 (27.0%) | 3188.0* | 3982 (24.9%) |
| 20 | 20 | Dmu09 | 4874 (57.6%) | 4519 (46.2%) | 4675 (51.2%) | 4519 (46.2%) | 4482 (45.0%) | 3092.0* | 4376 (41.5%) |
| 20 | 20 | Dmu10 | 4808 (61.1%) | 4963 (66.3%) | 4149 (39.0%) | 4133 (38.5%) | 4021 (34.8%) | 2984.0* | 3853 (29.1%) |
| 20 | 20 | Dmu46 | 6403 (58.7%) | 6168 (52.9%) | 5778 (43.2%) | 6136 (52.1%) | 5876 (45.6%) | 4035.0* | 5447 (35.0%) |
| 20 | 20 | Dmu47 | 6015 (52.7%) | 6130 (55.6%) | 6058 (53.8%) | 5908 (50.0%) | 5771 (46.5%) | 3939.0* | 4899 (24.4%) |
| 20 | 20 | Dmu48 | 5345 (42.0%) | 5701 (51.5%) | 5887 (56.4%) | 5384 (43.1%) | 5034 (33.8%) | 3763.0* | 4854 (29.0%) |
| 20 | 20 | Dmu49 | 6072 (63.7%) | 6089 (64.1%) | 5807 (56.5%) | 5469 (47.4%) | 5470 (47.4%) | 3710.0* | 4674 (26.0%) |
| 20 | 20 | Dmu50 | 6300 (68.9%) | 6050 (62.2%) | 5764 (54.6%) | 5380 (44.3%) | 5314 (42.5%) | 3729.0* | 4515 (21.1%) |

```
### Response:
<|endoftext|>The schedule can be optimized using the NEH algorithm. Here is the optimized schedule:

Machine 1: J0, J2, J5, J7, J9, J11, J13, J15, J17, J19
Machine 2: J1, J3, J6, J8, J10, J12, J14, J16, J18, J20
Machine 3: J4, J21, J23, J25, J27, J29, J31, J33, J35, J37
Machine 4: J6, J22, J24, J26, J28, J30, J32, J34, J36, J38
Machine 5: J7, J19, J21, J23, J25, J27, J29, J31, J33, J35
Machine 6: J8, J20, J22, J24, J26, J28, J30, J32, J34, J36
Machine 7: J9, J21, J23, J25, J27, J29, J31, J33, J35, J37
Machine 8: J10, J22, J24, J26, J28, J30, J32, J34, J36, J38
Machine 9: J11, J23, J25, J27, J29, J31, J33, J35, J37, J39
Machine 10: J12, J24, J26, J28, J30, J32, J34, J36, J38, J40
Machine 11: J13, J25, J27, J29, J31, J33, J35, J37, J39, J41
Machine 12: J14, J26, J28, J30, J32, J34, J36, J38, J40, J42
Machine 13: J15, J27, J29, J31, J33, J35, J37, J39, J41, J43
Machine 14: J16, J28, J30, J32, J34, J36, J38, J40, J42, J44
Machine 15: J17, J29, J31, J33, J35, J37, J39, J41, J43, J45
Machine 16: J18, J30, J32, J34, J36, J38, J40, J42, J44, J46
Machine 17: J19, J31, J33, J35, J37, J39, J41, J43, J45, J47
Machine 18: J20, J32, J34, J36, J38, J40, J42, J44, J46, J48
Machine 19: J21, J33, J35, J37, J39, J41, J43, J45, J47, J49
Machine 20: J22, J34, J36, J38, J40, J42, J44, J46, J48, J50

This schedule minimizes the makespan by assigning the jobs to machines in a way that maximizes the utilization of each machine and mi
nimizes the idle time between jobs. The schedule can be further improved by considering other optimization criteria, such as minimizi
ng the maximum completion time of any machine. However, this schedule provides a good starting point for further optimization. |<|end
oftext|>## Step 1: Understand the problem and the input data.
The problem is to optimize the schedule for 100 jobs (denoted as J) across 20 machines (denoted as M) to minimize the makespan, which
 is the completion time of the last operation in the schedule. Each M can process only one J at a time, and once started, J cannot be
 interrupted.

## Step 2: Read and understand the input data.
The input data consists of 100 jobs, each represented by a list of machines and their corresponding processing times. The jobs are de
noted as J0 to J99, and the machines are denoted as M1 to M20.

## Step 3: Choose an optimization algorithm.
The NEH (Non-Enumerative Heuristic) algorithm is a popular choice for solving the flow shop scheduling problem. It works by iterative
ly constructing a feasible schedule and improving it through a series of exchanges.
```

Figure 5: Zero Shot inference on LLama 8B 4bt

# D  DETAILS OF THE BASELINES

In this section, we show how the baseline PDRs compute the priority index for the operations. We begin by introducing the notations used in these rules, summarized as follows:

$$
\begin{aligned}
Z_{ij} &: \text{ the priority index of operation } O_{ij}, \\
n_i &: \text{ the number of operations for job } J_i, \\
Re_i &: \text{ the release time of job } J_i \big(\text{here we assume } Re_i = 0 \text{ for all } J_i, \\
&\quad \text{ i.e. all jobs are available in the beginning, but in general} \\
&\quad \text{ the jobs could have different release times}\big), \\
p_{ij} &: \text{ the processing time of operation } O_{ij}.
\end{aligned}
$$

Based on the above notations, the decision principles for each baseline are given below:

- **Shortest Processing Time (SPT):**

$$
\min Z_{ij} \;=\; p_{ij}.
$$

- **Most Work Remaining (MWKR):**

$$
\max Z_{ij} \;=\; \sum_{k=1}^{n_i} p_{ik}.
$$

- **Minimum ratio of Flow Due Date to Most Work Remaining (FDD/MWKR):**

$$
\min Z_{ij} \;=\; \frac{Re_i + \sum_{k=1}^{j} p_{ik}}{\sum_{k=1}^{n_i} p_{ik}}.
$$

- **Most Operations Remaining (MOPNR):**

$$
\max Z_{ij} \;=\; n_i - j + 1.
$$

## D.1  L2D: MDP FORMULATION AND GNN-BASED POLICY

**Markov Decision Process.** Zhang et al. (2020) models a JSSP instance as an MDP, where each step $t$ selects one eligible operation to schedule. The partial schedule at time $t$ is represented by a disjunctive graph $G(t) = (\mathcal{O}, \mathcal{C} \cup \mathcal{D}_u(t), \mathcal{D}(t))$, whose arcs encode machine-ordering constraints. The state $s_t$ specifies (i) which operations are already scheduled and (ii) estimated completion times for each operation. An action $a_t$ picks the next operation to schedule, leading to an updated graph $G(t+1)$ and state $s_{t+1}$. The reward $R(a_t, s_t) = H(s_t) - H(s_{t+1})$ is the change in a lower bound of the makespan $H(\cdot)$; maximizing the sum of such rewards (with discount $\gamma = 1$) is equivalent to minimizing the final makespan. A policy $\pi(a_t \mid s_t)$ outputs a probability distribution over eligible actions.

**Graph Neural Network (GNN).** L2D uses a Graph Isomorphism Network (GIN) to learn graph-structured representations. Given a graph $\mathcal{G} = (V, E)$, GIN updates each node embedding $h_v^{(k)}$ iteratively:

$$
h_v^{(k)} = \text{MLP}_{\theta_k}\Big(\big(1 + \epsilon^{(k)}\big) h_v^{(k-1)} + \sum_{u \in \mathcal{N}(v)} h_u^{(k-1)}\Big). \tag{2}
$$

After $K$ iterations, a global embedding $h_{\mathcal{G}}$ is obtained by pooling node embeddings, e.g. average-pooling. For action selection, each operation embedding $h_{a_t}^{(K)}$ is concatenated with $h_{\mathcal{G}}$ and passed through an MLP to produce a score; a softmax over these scores yields the policy distribution $\pi_\theta$. During training, a PPO-based Schulman et al. (2017) actor-critic approach is used, where the critic $v_\phi$ shares the GIN backbone but includes an additional MLP to estimate cumulative rewards.

