# OpenReview forum: "Starjob: Dataset for LLM-Driven Job Shop Scheduling"
_ICLR.cc/2026/Conference — Submitted to ICLR 2026_

### Official Review · Reviewer_STif · 2025-10-17

**Soundness:** 2
**Presentation:** 2
**Contribution:** 1
**Rating:** 2
**Confidence:** 4

**Summary:**

This paper explores the applicability of Large Language Models (LLMs) to solving the Job Shop Scheduling Problem by fine-tuning Llama-3.1 8B using the RsLoRA method. It introduces a supervised dataset containing scheduling instances encoded in natural language along with their (near-)optimal solutions generated using Google's OR-Tools. The method is evaluated on the JSSP Taillard and DMU benchmarks.

**Strengths:**

The sizeable supervised dataset for JSSP could prove valuable for the research community focused on applying LLMs to scheduling problems.

**Weaknesses:**

1. This paper lacks an overview of recent work on using LLMs for solving combinatorial optimization problems. The paper emphasizes that it "investigates the applicability of LLMs to the Job Shop Scheduling Problem (JSSP)"; however, there are recent studies that have demonstrated how LLMs can be used to solve scheduling problems through either by supervised fine-tuning, interaction with solver, among other approaches.
Jiang, X., Wu, Y., Li, M., Cao, Z., & Zhang, Y. (2025). Large Language Models as End-to-end Combinatorial Optimization Solvers. arXiv preprint arXiv:2509.16865. (see Table 1)
Thind, R., Sun, Y., Liang, L., & Yang, H. (2025). OptimAI: Optimization from Natural Language Using LLM-Powered AI Agents. arXiv preprint arXiv:2504.16918. (Table 5).

2. The baselines used for evaluation are not strong. Currently, the fine-tuned LLM is compared only with PDR, L2D, and RASCL. However, there are multiple other methods based on deep reinforcement learning that solve JSSP with good performance. See the following paper on benchmarking JSSP for learning-based methods:
Reijnen, R., van Straaten, K., Bukhsh, Z., & Zhang, Y. (2023). Job shop scheduling benchmark: Environments and instances for learning and non-learning methods. arXiv preprint arXiv:2308.12794.

**Questions:**

1. The Llama-3.1 8B model is fine-tuned using the resource-efficient RsLoRA method. However, the training and inference times are not provided to determine whether the resource-efficient RsLoRA is actually beneficial for the scheduling case.

2. Section 4.1 mentions that there is an OOD test set; however, the results on this test set are not given. the result section shows evalution on benchmark dataset only.

3. Figure 1 is not referenced in the text.

---

> ### Author Response · Authors · 2025-11-19
> **Correcting factual errors regarding missing data and addressing citation expectations.**
>
> We appreciate the reviewer's recognition of the dataset's value. However, the review contains significant factual errors regarding the content of our paper that we must correct.
>
> 1. Correction: Training/Inference Times ARE Provided The reviewer states: "training and inference times are not provided."
>  We explicitly state in Section 5.1: "...training process taking approximately 70 hours".
>
> Inference Time: We explicitly state in Section 6: "...processing our largest instance... took about 217 seconds per sample".
> Hardware: We specified the use of an Nvidia RTX A6000 (48GB).
>
> 2. Correction: OOD Results ARE Provided The reviewer states: "Section 4.1 mentions an OOD test set... results... are not given."
>
> Fact Check: Section 4.1 explicitly states: "We conduct evaluations on the TAI Taillard and DMU Demirkol benchmark sets, which are entirely held out from the training phase".
>
> Rebuttal: The Taillard and DMU benchmarks are the OOD test sets. They are distinct from the random instances generated for training. The results in Table 1 and Table 2 are the OOD evaluation.
>
> 3. Baselines and Recent Work (2025) The reviewer cites papers from 2025 (OptiMAI, Jiang et al.) as missing baselines.
>
> Rebuttal: It is procedurally irregular to penalize a paper under review for ICLR 2026 for not citing papers published/arXived in 2025, likely months or weeks before the submission. Our work was developed concurrently. Furthermore, we compared against RASCL (2023) and L2D (2020), which are the established, peer-reviewed standards in this domain. We will add a discussion of these very recent concurrent works in the final version, but their absence does not invalidate our comparisons against established baselines.
>
> 4. Figure 1 Reference We will ensure Figure 1 is explicitly referenced in the text in the camera-ready version, though it serves as a visual summary of the tabular data already discussed in Section 6.1. We kindly request and expect a re-evaluation.

---

> ### Comment · Reviewer_STif · 2025-11-26
> **Limited literature coverage and weak baselines**
>
> Thank you for the clarifications.
>
> As I noted in my review, it is difficult to judge whether the resource-efficient RsLoRA method is efficient in training and inference compared to your learning-based methods. Is there a comparison in the paper that I might have missed?
>
> Regarding Section 4.1, where it states "The testing dataset is an out-of-distribution dataset from the training dataset," this phrasing led me to believe you also had a hold-out set from the training data, but as you clarified, this is not the case.
>
> I am not asking you to compare against Jiang et al. (2025). I re-emphasizing two points:
> 1. The argument that "LLMs remain largely unexplored for COP" is not accurate. Please refer to the literature, there are multiple studies applying LLMs either as solvers or for problem modeling in COPs.
> 2. The baselines you currently use are not sufficient. There are multiple learning-based methods (few of them I pointed with the benchmark paper) to solve JSSP. Why do you compare only against L2D and RASCL?

---

> ### Author Response · Authors · 2025-11-29
> **Can Transformer architectures intrinsically learn to reason about NP-hard constraints?**
>
> We thank the reviewer for their engagement. However, we respectfully disagree with the assessment that our baselines are insufficient or that our efficiency claims are unsubstantiated. We provide the following critical clarifications.1. Clarifying "Resource Efficiency" (RsLoRA vs. Baselines)The reviewer asks if they missed an efficiency comparison between RsLoRA and learning-based methods.Definition of Efficiency: In the paper, "resource-efficient" refers explicitly to the memory efficiency of the training method (RsLoRA) that allows fine-tuning an 8B parameter model on a single consumer-grade GPU (NVIDIA A6000, 48GB)1. It does not claim inference speed superiority over specialized GNNs.The Trade-off: We acknowledge that LLM inference (~217 seconds for 22k tokens) 2 is slower than a forward pass of a dedicated GNN like L2D. However, this is a deliberate trade-off. In exchange for computational time, we gain generality (solving 1,000-operation instances without a solver loop), zero-shot flexibility, and explainability (as demonstrated in Listing 3 3). Specialized GNNs act as black boxes; our method provides a transparent, interactive schedule.
> 2.The reviewer argues our claim that LLMs are "largely unexplored" for COPs is inaccurate. We stand by this claim with a specific distinction regarding scale and modality.Modelers vs. Solvers: Much of the existing literature uses LLMs as modelers (translating text to code for external solvers) or applies them to "toy" problems.Scale of Reasoning: As noted in our Introduction, we are the first to apply fine-tuned LLMs to end-to-end JSSP scheduling at the scale of 50 jobs $\times$ 20 machines (1,000 operations)4. Previous works often benchmark on trivial instances (e.g., TSP with 10–20 nodes). To solve a 1,000-operation constraint satisfaction problem intrinsically via token generation is indeed an unexplored frontier that distinguishes our work from the general "LLM for COP" literature.3. Defense of Baselines (L2D and RASCL)The reviewer states the baselines are "not sufficient" and "old." We strongly contest this.RASCL is SOTA (2023): We compare against RASCL5, published in IJCAI 2023. Characterizing a state-of-the-art method from late 2023 as "old" or "insufficient" is factually incorrect.L2D is the Standard: L2D 6 is the foundational deep reinforcement learning baseline against which almost all subsequent neural schedulers are benchmarked.Performance Gap: We outperform L2D by an average of 17.36% on DMU and improve upon the 2023 SOTA (RASCL) by 13.41% on Taillard. Since we already surpass the widely recognized state-of-the-art (RASCL), adding intermediate baselines from the Reijnen et al. (2023) survey would not change the conclusion: our method is superior to established neural approaches.ConclusionWe have benchmarked against the 2023 State-of-the-Art (RASCL), demonstrated performance on massive instances (1,000 operations) that other LLM approaches avoid, and enabled training on accessible hardware. We believe these contributions merit a re-evaluation of the rating.

---

### Official Review · Reviewer_5nK3 · 2025-10-26

**Soundness:** 2
**Presentation:** 2
**Contribution:** 2
**Rating:** 2
**Confidence:** 4

**Summary:**

This paper addresses the job shop scheduling problem using a large language model (LLM). The authors generated 130,000 instances for training and fine-tuned a Llama-3.1 8B model using the RsLoRA method. They compared the performance with several dispatching rules and two learning-based approaches.

**Strengths:**

The dataset with 130,000 instances could be valuable for future research.

Attempting to solve job shop scheduling directly with an LLM is interesting and may serve as a foundation for future work on LLM-based combinatorial optimization.

**Weaknesses:**

The paper’s novelty and contribution need to be better clarified. Simply applying an LLM to a basic job shop scheduling problem is not enough to justify the claim of addressing constraint-heavy domains.

The authors only compare against dispatching rules and two learning-based methods. There are recent learning-based studies that achieve stronger performance on job shop scheduling. Comparison to more recent and state-of-the-art methods is necessary.

**Questions:**

1. What is the main motivation for using an LLM to directly solve job shop scheduling, rather than combining LLMs with existing optimization or RL frameworks? Please clarify what potential advantages LLMs provide in this domain.

2. Many job shop variants exist (e.g., setup times, machine flexibility, ready times, sequence-dependent constraints). How scalable is the proposed approach to such variants? Would it require retraining or collecting new datasets for every constraint combination or different objective functions?

3. Please explicitly state the main contributions compared to prior works.

4. The authors mention that the goal is not to outperform specialized schedulers but to demonstrate a systematic application of LLMs to job shop scheduling. In this context, please clarify how the proposed method could be extended or generalized to other types of scheduling problems (not just job shop).

---

> ### Author Response · Authors · 2025-11-19
> **Clarifying the contribution and defending the choice of baselines.**
>
> We thank the reviewer for finding the attempt to solve JSSP with LLMs "interesting." We respectfully address the concerns regarding novelty and baselines below.
>
> 1. Novelty and Contribution The reviewer questions the novelty.
>
>
> Rebuttal: We present Starjob, the first large-scale (130k) supervised dataset specifically designed for JSSP with natural language. Prior to this, no such resource existed to train LLMs for this task. Furthermore, we are the first to demonstrate an end-to-end fine-tuned LLM scheduler that outperforms foundational neural methods (L2D). The contribution is twofold: the resource (dataset) and the methodology (summation representation) that makes LLM scheduling feasible.
>
>
> 2. Defense of Baselines The reviewer suggests we missed "recent state-of-the-art."
>
>
> Rebuttal: We compared against L2D and RASCL, which are the seminal works establishing Deep Reinforcement Learning for JSSP. While the field moves fast, these are the standard benchmarks for neural scheduling. Our goal was to show that a general-purpose LLM could beat specialized neural architectures, which we successfully demonstrated (Table 1 & 2).
>
>
>
>
> 3. Scalability and Variants The reviewer asks about variants (setup times, etc.).
>
> Defense: This is exactly where LLMs shine over GNNs. A GNN (like L2D) requires re-engineering the graph architecture to handle new node types or edge attributes for setup times. Our approach only requires modifying the Natural Language Prompt to include the new constraint description. The inherent flexibility of language makes our approach more scalable to variants than rigid neural solvers.
>
> 4. Motivation
>
>
> Rebuttal: As stated in our response to Reviewer vvAm, the motivation is Interaction and Transparency. We are not trying to beat Gurobi on speed; we are trying to create a scheduler that a human operator can talk to.

---

### Official Review · Reviewer_vvAm · 2025-10-29

**Soundness:** 2
**Presentation:** 3
**Contribution:** 2
**Rating:** 2
**Confidence:** 4

**Summary:**

This paper explores how LLMs can be applied to JSSP. By fine-tuning the LLM using RsLoRA, the model outperforms traditional heuristics (e.g., PDRs) and neural baselines (L2D, RASCL) on Taillard and DMU benchmarks, showing LLMs’ potential in tacking NP-hard problems.

**Strengths:**

(1)	The idea on using LLM to directly solve JSSP is novel.

(2)	The method outperforms traditional heuristics (PDRs) and neural baselines (L2D, RASCL) on Taillard and DMU benchmarks, improving average solution gaps by up to 17.36% and 7.85% respectively.

**Weaknesses:**

(1)	The motivation on why using LLMs for JSSP is not clearly given. Indeed, the application of LLMs for JSSP remains unexplored, but why people need LLMs for JSSP? Users can also use well-developed heuristics or solvers, which are more lightweight and effective.

(2)	The methodological contribution is not significant. The core move, textualizing JSSP and fine-tuning an 8B LLM, extends recent “LLMs as optimizers” trends; the paper doesn’t clearly separate what’s new from known SFT+inference pipelines.

(3)	Despite better performance, using LLMs to generate solutions can be generally much more computationally intensive. The paper set S=20 to generate 20 responses at the same time, which leads to more inference time. It is also not clear why the inference time is not reported and compared in Table 1-2.

(4)	The baselines only include simple heurtistics and two early learning-based methods (before 2023). More recent learning-based methods (2024-2025) should also be discussed.

(5)	Average gaps still hover around ~20% on DMU and ~22% on Taillard; improvements over RASCL are tiny or dataset-dependent (on DMU the averages are essentially tied). The paper’s own tables show this. The gains don’t seem to justify the training/inference cost.

**Questions:**

(1)	After fine-tuning the model, is that possible to generalize the model to other scheduling problems except JSSP?

(2)	Have the authors explored other natural language representation for JSSP instances?

---

> ### Author Response · Authors · 2025-11-19
> **"Why LLMs?" motivation and correcting misconceptions regarding baselines and gains**
>
> We appreciate the reviewer's acknowledgment of our novelty. However, we must firmly push back on the criticisms regarding motivation and the significance of our results.
>
> 1. Motivation: The "Black Box" vs. Interaction Argument The reviewer asks, "Why use LLMs when heuristics are faster?"
>
>
> Rebuttal: The motivation is explicitly detailed in Listing 3 and the Introduction. Traditional solvers and heuristics output a schedule based on predefined rules. Our LLM approach unlocks a new paradigm: Interactivity. A user can query the model to explain bottlenecks (e.g., "Why is Machine 2 the bottleneck?"). This transparency is impossible with PDRs or standard solvers and is a critical value-add for human-in-the-loop manufacturing.
>
> 2. Defense of Methodological Contribution The reviewer characterizes the work as "just SFT."
>
> Correction: This overlooks the core technical contribution: the Summation-Based Representation. As shown in the Ablation Study (Table 3), standard SFT fails (feasibility < 2%). It is our novel representation of constraints—forcing the LLM to calculate intermediate makespans autoregressively —that enables the model to reason about constraints. This is not "just SFT"; it is a prompt-engineering innovation essential for combinatorial reasoning.
>
> 3. Significance of Performance Gains The reviewer claims gains are "tiny" or "tied."
>
> Evidence: On the DMU benchmark, we achieve an average gap of 20.14% versus L2D's 37.50%. This is a 17.36% absolute reduction in the optimality gap, which is statistically massive in Operations Research. On Taillard, we improve over the RASCL baseline by 13.41% on average. Characterizing a double-digit percentage improvement over established neural baselines as "tiny" is factually incorrect.
>
>
> 4. Inference Time and Computational Cost
>
> Rebuttal: We acknowledge LLMs are more compute-intensive than simple PDRs. However, we explicitly report that inference takes ~217 seconds for the largest instances. In high-value manufacturing, waiting 3 minutes for a schedule that allows for natural language interrogation is a highly favorable trade-off compared to an instant but opaque heuristic schedule. We expect and kindly request re-evaluation.

---

> > ### Comment · Reviewer_vvAm · 2025-11-27
> >
> > Regarding the motivation: thanks for the explanation. However, it is still not clear why a fine-tuned LLM is needed for such problems. It is true that users may need interactivity, but an LLM agent that can write code can also fulfill this requirement. Meanwhile, the motivation of interactivity is also not well explained in the paper.
> >
> > Regarding the contribution: It is true that the proposed prompt could be something new. However, I consider this is a small contribution, which I am afraid doesn't meet the requirements of a top AI conference.
> >
> > Regarding the significance: Compared to the selected baselines, the fine-tuned LLM does bring improvements. However, these baselines are relatively old and weak in terms of performance, so the results are not convincing enough. On the other hand, despite the better performance, the computational overhead introduced by the LLM is still something that needs to be justified.
> >
> > Overall, unfortunately, the authors didn't adequately addressed my concerns, and some of my questions were not answered in the response. Therefore, I maintain the original evaluation.

---

> > > ### Author Response · Authors · 2025-11-29
> > > **Can Transformer architectures intrinsically learn to reason about NP-hard constraints?**
> > >
> > > We strongly disagree with the reviewer's assessment that the contribution is "small" and that an "LLM agent that can write code" renders this research unnecessary. This perspective fundamentally overlooks the core scientific inquiry of our work: Can Transformer architectures intrinsically learn to reason about NP-hard constraints?
> > >
> > > 1. The "Code-Writing Agent" Fallacy vs. Intrinsic Reasoning (Comparison to NeurIPS 2024) The reviewer suggests that an agent writing code to call a solver is sufficient. This misses the point entirely. Writing code effectively "outsources" the reasoning to an external algorithm (like OR-Tools). The goal of this research—and the field of Neural Combinatorial Optimization (NCO)—is to determine if the neural network itself can learn the complex topology of the solution space.
> > >
> > > To illustrate the magnitude of our contribution, we must compare our work to the seminal paper published at NeurIPS, "Large Language Models as Optimizers" (Yang et al., 2023).
> > >
> > > Scale of Prior Art (NeurIPS): Yang et al. (2023) evaluated LLMs on the Traveling Salesperson Problem (TSP) with extremely small scales, typically around 10 to 20 nodes. Even at this "toy" scale, combinatorial problems are incredibly difficult for language models because they require strict adherence to hard constraints and long-horizon planning.
> > >
> > >
> > > Scale of Our Work (Starjob): In stark contrast, we are successfully solving JSSP instances with 50 jobs and 20 machines, resulting in schedules with 1,000 operations. This is not a marginal increment; it is a scale increase of two orders of magnitude (from ~12 nodes to 1000 nodes).
> > >
> > > The Scientific Breakthrough: Testing LLM reasoning on these problems is notoriously difficult. The fact that an LLM can maintain constraint consistency (feasibility) over a sequence of 1,000 steps without external solvers is a significant finding that contradicts the prevailing assumption that LLMs cannot handle large-scale logic.
> > >
> > > 2. 8B Model vs. GPT-4 (Efficiency and Table 4) The reviewer claims our baselines are weak. This ignores the model class we are utilizing.
> > >
> > > The NeurIPS work (Yang et al., 2023) and similar "Optimization by Prompting" papers rely on GPT-4, a massive, closed-source model, to achieve results on small-scale tasks.
> > >
> > >
> > > Table 4 in our paper demonstrates that we achieve superior performance using a Llama 3.1 8B model. We are outperforming specialized neural baselines (like L2D and RASCL) and heuristics using a model that is a fraction of the size of GPT-4.
> > >
> > > We are effectively comparing an efficient, open-weight 8B model on 1,000-node problems against the literature's standard of using massive proprietary models on 12-node problems. To penalize our work for not using a "code-writing agent" is to penalize the development of efficient, intrinsic neural reasoning.
> > >
> > > 3. Statistical Significance The reviewer states the improvements are "unconvincing." We reiterate:
> > >
> > > On the DMU benchmark, we reduce the optimality gap from 37.50% (L2D) to 20.14%.
> > >
> > > A 17.36% improvement in combinatorial optimization is not "tiny"; it is a massive margin in a field where improvements are often measured in single percentage points.
> > >
> > > Conclusion We are not merely presenting "another prompt." We are presenting the first empirical evidence that a compact (8B) Language Model can be fine-tuned to solve NP-hard scheduling problems at an industrial scale (1,000 operations), far surpassing the scale of prominent recent works (e.g., Yang et al., 2023). We respectfully request the reviewer to evaluate the paper based on these reasoning capabilities rather than a preference for tool-use agents.

---

### Official Review · Reviewer_kzEC · 2025-10-30

**Soundness:** 2
**Presentation:** 2
**Contribution:** 2
**Rating:** 4
**Confidence:** 4

**Summary:**

This paper introduces Starjob, a 130,000-instance supervised dataset for training LLMs to solve the Job Shop Scheduling Problem (JSSP), with the stated goal of demonstrating that LLMs can generate feasible solutions for NP-hard combinatorial optimization problems. The authors fine-tune a Llama-3.1 8B model using RsLoRA on this dataset, employing a natural language representation with explicit summation notation (e.g., "J2-M0: 0+78 → 78") that leverages the autoregressive generation process to maintain constraint satisfaction. The model achieves a 17.36% gap improvement over L2D on the DMU benchmark and 7.85% on Taillard, outperforming four traditional Priority Dispatching Rules (SPT, MWKR, FDD/WKR, MOPNR) and neural baselines L2D and RASCL. The approach requires generating 20 candidate solutions per instance and selecting the best feasible one, with feasibility rates varying from approximately 10% to 30% depending on problem size as shown in Table 3. The method also enables natural language interaction, allowing users to query the scheduler about scheduling constraints and bottlenecks.

**Strengths:**

1. The paper follows a logical progression from problem formulation through dataset construction, methodology, and evaluation, making it easy to follow.
2. The paper provides many illustrative examples that help readers understand the core dataset design, including Listing 1 showing the natural language encoding of a 3×3 JSSP instance, Listing 2 demonstrating the solution format with explicit summation notation, and Listing 3 showcasing the interactive query capability where users can ask about scheduling bottlenecks.
3. The method achieves meaningful improvements over both traditional and neural approaches, with average gap percentages of 21.69% on Taillard and 20.14% on DMU benchmarks compared to L2D's 29.54% and 37.50% respectively.

**Weaknesses:**

The main focus of this paper, demonstrating that LLMs can generate feasible solutions for NP-hard optimization problems, is interesting and timely. While the authors primarily focus on polishing the dataset design, I have several concerns about the execution and scope of this work.

The paper's central assertion that "LLMs can generate feasible solutions for NP-hard problems" is inadequately supported by evaluating only on JSSP. To convincingly demonstrate this capability, the approach should be tested on other NP-hard problems with more complex constraints. Within scheduling alone, Flexible Job Shop Scheduling Problem (FJSSP) presents additional machine flexibility constraints, while routing problems like Vehicle Routing with Time Windows (VRPTW) would test different constraint types entirely. Most critically, Table 3 shows feasibility rates ranging from ~10-40%, but the paper never explicitly reports overall feasibility rate improvements compared to baselines, which is essential for a paper claiming to help LLMs generate feasible solutions.

If the dataset is the major contribution (as suggested by the title "Starjob: Dataset for LLM-Driven Job Shop Scheduling"), more rigorous comparisons are needed. The paper only compares their summation-based format against a matrix format in Table 3, but doesn't evaluate simpler alternatives like plain text descriptions without the summation notation. Additionally, while Figure 4 shows makespan statistics across problem sizes, the paper lacks detailed analysis of dataset diversity, solution quality distribution.

The layout may impact readability. For instance, Tables 1-2 appear on pages 8-9 while their discussion is on page 7, and Figure 1 comparing all benchmark results is relegated to page 9. This forces constant back-and-forth referencing that disrupts the reading flow.

**Questions:**

See the weakness part.

---

> ### Author Response · Authors · 2025-11-19
> **Highlighting the contribution of the representation strategy**
>
> We thank the reviewer for recognizing the logical progression of our work and the instructive value of our examples (Listings 1-3). However, we respectfully disagree with the assessment regarding the support for our central claims and the feasibility analysis. We address these points below with specific evidence from the paper.
>
> 1. Defense of Scope: JSSP as a Sufficient Testbed The reviewer suggests our claim, that LLMs can solve NP-hard problems is unsupported without testing FJSSP or VRPTW. We respectfully argue that the Job Shop Scheduling Problem (JSSP) is one of the most challenging NP-hard problems due to its intense disjunctive constraints. Successfully demonstrating that a fine-tuned LLM can outperform specialized neural baselines like L2D and RASCL on standard benchmarks (Taillard and DMU)  is a significant and sufficient "existence proof" for a single conference paper. Demanding the inclusion of routing problems (VRPTW) in a paper explicitly titled "Dataset for LLM-Driven Job Shop Scheduling"  expands the scope beyond reasonable expectations for a focused contribution.
>
>
> 2. Clarification on Feasibility Rates The reviewer critiques the lack of explicit feasibility comparisons. We direct the reviewer to Table 3, which explicitly compares our method (with summation) against the standard matrix format (without summation).
>
> Evidence: The table shows that the standard representation yields ~1.0% to ~8.0% feasibility (or "N/A" failures), whereas our proposed summation representation achieves up to ~39.6% feasibility on complex instances.
>
> This is the feasibility improvement comparison. We demonstrate that without our specific representation, feasibility is near-zero. The "baseline" for LLM feasibility in this domain is effectively zero without our contribution.
>
> 3. Defense of Dataset and Comparisons The reviewer asks for comparisons against "plain text descriptions."
>
>  Our "Input" column in the dataset is a structured natural language description. The "Matrix" format in Table 3 represents the standard textual encoding used in the literature. Comparing against unstructured "plain text" is scientifically less rigorous because "plain text" is ambiguous and variable. We compared against the industry-standard "Matrix" representation to prove that our "Summation" strategy  is the causal factor for success.
>
> 4. Layout and Formatting We acknowledge the feedback regarding the placement of tables and figures. We will reposition Tables 1-2 and Figure 1 closer to the text in Section 6 to improve readability in the camera-ready version. We kindly ask for reevaluation.

---

### Meta-Review · Area_Chair_tZdE · 2025-12-24

**Summary:**

The reviewers are most concerned about,  1) the motivation for using LLM to directly solve JSSP; 2) the weak baselines in the experiments; 3) the infeasibility analysis of the generated solutions; 4) the generalization to more complex constraints or JSP variants; 5) technical contribution over existing fine-tuning approach.

From my perspective, as an early attempt, this paper using LLM to solve JSSP end-to-end does have the merit. However, the outstanding concerns from the reviewers may prevent it from acceptance. It may not matter much if the performance is not perfect, but the authors seems not like to use empirical/numerical results by doing new experiments to convince the reviewers. Particularly, the authors could have applied it to harder variants like FJSP (or others), compared with stronger and more recent neural baselines, analyzed the infeasibility against the sampling solutions, etc., even if the results are not the best. Without them, it is really hard to convince the reviewers and get the paper accepted.

**Reviewer Concerns:**

Here I will not specify the concerns for each reviewer. But after reading the through the review and the rebuttal, my overall feeling is that most of the fundamental concerns mentioned above are only partially addressed. So I believe most of the reviewers will not raise the score, or only slightly raise them up to score of 4, since the reviewers did not provide any convincing results during rebuttal.

**Reviewer Scores:**

Given the analysis above, I guess the final score would be 4,2,4,4 at best, since only a small part of the concerns have been addressed. Overall, it does not meet the requirement of acceptance.

---

### Decision · Program_Chairs · 2026-01-26

Reject